# Coupling Effects Analysis and Suppression in a Highly Integrated Ka-Band Receiver Front-End MMIC for a Passive Millimeter-Wave Imager System

**DOI:** 10.3390/s22155695

**Published:** 2022-07-29

**Authors:** Xi Chen, Anyong Hu, Jianhao Gong, Amjad Altaf, Jungang Miao

**Affiliations:** 1School of Electronic and Information Engineering, Beihang University, Beijing 100191, China; chenxi0913@buaa.edu.cn (X.C.); gongjh@buaa.edu.cn (J.G.); jmiaobremen@buaa.edu.cn (J.M.); 2Center for Excellence in Applied Sciences & Technology (CESAT), Islamabad 44800, Pakistan; amjad_altaf@buaa.edu.cn

**Keywords:** coupling effects, image rejection ratio, MMIC, passive millimeter-wave imaging, receiver

## Abstract

This paper presents the coupling effects analysis and suppression of a highly integrated receiver front-end MMIC for a passive millimeter-wave imager system. The receiver MMIC consists of a low-noise amplifier, double-balanced image-reject mixer, frequency quadrupler, and analog phase shifter. In order to integrate these devices into a compact single chip without affecting the core performance, coupling problems need to be solved. We analyze the influence of coupling effects on the image rejection ratio, and propose corresponding solutions for three different coupling paths. (1) The coupling in the LO-RF path of the mixer is solved by designing a double-balanced mixer with high isolation characteristics. (2) The coupling between the LO chain and the LNA from space and dielectric is suppressed by optimizing the two main transmission lines spacing and adding isolation vias. (3) The coupling caused by the line crossing is restrained by designing a differential line crossover structure. The design and implementation of the MMIC are based on 0.15 µm GaAs pHEMT process. The receiver chip has 6.1~8.7 dB conversion gain in 32~36 GHz, less than 3.5 dB of noise figure, and more than 35 dB of image rejection ratio. The measurement results show that the receiver MMIC is especially suitable for high-sensitivity passive millimeter-wave imaging systems.

## 1. Introduction

Passive millimeter wave imaging (PMMI) is an effective technique for detecting concealed weapons or contraband using high-sensitivity radiometers, and it has been widely used in the field of human security screening [1,2,3]. In recent years, to obtain a better spatial resolution, radiometric sensitivity, or imaging rate, the systems are developing toward large-scale array [4]. The application background of this paper is a 1024-channel PMMI system (BHU-1024) [5], which uses phased array beam scanning and interferometry to obtain two-dimensional images. In order to realize such a large-scale array, as the core component of the radiometer, the receiver front-end with the characteristics of high integration, low power consumption, and low cost is very much desired.

Multifunctional Monolithic Microwave Integrated Circuit (MMIC) is one of the suitable implementation methods. Compared with the CMOS process [6,7,8,9], MMIC based on the GaAs process has advantages in low noise characteristics, which is more suitable for systems with high-sensitivity requirements. GaAs-based integrated receiver MMICs have already reported some application cases, such as a Ka-Band sub-harmonic image-reject down-converter MMIC in [10] only integrates low noise amplifier (LNA) and image-reject mixer (IRM), and the return loss of the LO port is lower than 9 dB. Although the integration level of the down-conversion chip in [11] and the receiver chip in [12] is improved, their image rejection ratio (IRR) is 10~15 dB. In addition, according to the single-ended mixer structure similar to [13], the isolation between LO and RF is approximately 10 dB. The above-mentioned MMICs perform well in radar or communication fields, but their low image rejection will deteriorate the noise characteristics of small signals in the radiometer and fail to meet the requirements of PMMI systems. In addition, in order to implement a highly integrated MMIC, the inevitable coupling problems in the circuit also need to be analyzed and solved.

In this paper, a highly integrated receiver front-end MMIC is designed and implemented specially for PMMI system applications. The major contributions are summarized as follows:(1)The influence of various potential coupling effects in the integrated MMIC on IRR is quantitatively analyzed, and the design requirements are put forward accordingly;(2)Corresponding solutions are proposed for three coupling paths: By designing a double-balanced mixer (DBM) with high isolation characteristics, the mutual coupling in the RF-LO path is less than −34 dB; By tuning the distance between the main transmission path of LNA and LO chain, while adding isolation vias, the LO coupling from the space and dielectric is suppressed to −29 dB; By using differential lines to design the crossover structure, the coupling of the LO to the mixing unit is −25 dB, and the coupling of LO to the IF port is reduced to −40 dB by the 180° reverse superposition of balun;(3)Compared with the reported receiver MMICs based on similar processes, the MMIC proposed in this paper not only has a higher integration level, but also has advantages in its low noise characteristics (Noise figure 3~3.5 dB, IRR greater than 35 dB) and low power consumption (LO drive power −15 dBm, DC power consumption 330 mW). This receiver MMIC is very suitable for a high-sensitivity large-scale array PMMI system.

## 2. Integrated Receiver Front-End MMIC Overview and Coupling Analysis

### 2.1. Integrated Ka-Band Receiver Front-End MMIC Overview

According to the application requirements of the BHU-1024 PMMI system, the receiver is a down-conversion structure with a phase-shifting function. The basic architecture of the circuit consists of an LNA, an IRM, a frequency quadrupler (×4), and an analog phase shifter, as shown in Figure 1.

In this design, the RF operating frequency range is 32~36 GHz, and the IF output is 4~8 GHz by mixing with the LO signal of 40 GHz. The first stage LNA provides adequate gain and a low noise figure. The IRM is used to suppress the unwanted 44~48 GHz image frequency interference in the system. In addition to the quadrupler, amplifiers and band-pass filters (BPF) are cascaded in the LO chain to provide sufficient driving power for the mixer and to suppress harmonics. Placing the analog phase shifter at the front stage of the quadrupler in the LO chain has the following advantages: (1) The broadband phase shift in the RF range can be achieved through a narrow-band design; (2) It is easier to meet the phase shift range greater than 360° and high phase shift accuracy; and (3) During phase shifting, its amplitude change does not affect the RF passband characteristics.

### 2.2. Analysis of Potential Coupling Paths and Coupling Effects on IRR

When multiple single-function chips are integrated, the coupling effects will affect the original performances, so it is necessary to analyze potential coupling effects. Figure 1 shows the circuit connection relationship and signal direction from which it can be seen that α, β, and γ are three different coupling paths.

In the IRM, the RF and LO signals enter into two symmetrical mixing units through a Lange coupler and a power divider, respectively. If the port isolation of the mixing unit is low, mutual coupling α will occur between the RF and the LO paths. When designing an IRM separately, since each port is connected to a matched load, even if the coupling exists, it only means that the isolation between the RF and the LO ports is low, and other performances will not be affected. However, when the IRM is applied to an integrated chip, due to the imperfect interconnection matching, the mutual coupling signal will be reflected into the mixing units again and superimposed with the original signal, thus affecting the imbalance of in-phase and quadrature (I/Q) paths.

Due to the parallel layout of the 40 GHz amplifier and LNA, the coupling of a 40 GHz LO signal from space or dielectric to LNA can occur at multiple locations, such as the input port, intermediate stage, or output port of the LNA. Since the RF and the LO frequencies are similar, the LO coupling signal can still be amplified by LNA and transmitted to the IRM. So, the coupling β is defined as the LO coupled signal transmitted to the IRM input port, which includes not only the LO signal directly coupled to the IRM, but also the coupling signal amplified by LNA. The LO coupling signal entering from the RF port of the IRM will also deteriorate the imbalance.

In order to make the IF I/Q path symmetrical and ensure output from the same side, the LO transmission line and the IFQ path are crossed, and an air bridge is usually used to realize the crossover in GaAs MMICs. For high-frequency signals, coupling is easy to occur at the intersection position of lines, which we define as coupling γ. The LO coupling signal will be transmitted to the mixing unit on the Q path of the IRM, affecting imbalance. The coupling signal can also be transmitted to the IF output port, causing the high-power LO signal to leak into the off-chip IF chain.

Through the above preliminary analysis, all of these three kinds of coupling will affect the imbalance of the I/Q path of the integrated chip and then deteriorate the IRR. The relationship between the IRR and I/Q imbalance can be expressed by the following [14]:(1)$$\mathrm{IRR}=-10log\left[\frac{1-2\sqrt{G}cos\left(\theta \right)+G}{1+2\sqrt{G}cos\left(\theta \right)+G}\right]$$
where *θ* and *G* are the imbalance of phase and amplitude, respectively, and the calculation results are shown in Figure 2. When the system requires the IRR to be >20 dB, it can be seen from the figure that a series of amplitude and phase values can meet the requirements. Using one of the typical values as a reference, the design goal of the amplitude imbalance between I/Q paths is less than 1.4 dB, and the phase imbalance is less than 8°. When the mixer circuit structure is symmetrical, the quadrature Lange coupler mainly determines the imbalance of the I/Q paths. Taking the design in this paper as an example, the Lange coupler needs to cover the RF frequency of 32~36 GHz and the image frequency of 44~48 GHz. The simulation result of the imbalance is approximately 0.6 dB and 2.5°, as shown in Figure 3. That is to say, the imbalance introduced by coupling effects should be less than 0.8 dB and 5.5°.

The three coupling paths are modeled and calculated, respectively. The diagram of mutual coupling in the RF-LO path is shown in Figure 4. Parts of the $V_{RF1}$ and $V_{RF2}$ are coupled to the LO path and reach the output end of the BPF through the power divider. The RF and image frequencies are out-of-band for the 40 GHz BPF and usually do not match well, which means that the coupled RF signal is synthesized and reflected at the output of the BPF and enters the two mixing units again. In this case, the amplitude and the phase imbalance of the I/Q paths change, which can be expressed as:
(2)$$V_{RFI\_\alpha }=\frac{1}{\sqrt{2}}V_{RF}e^{j(\varphi_{RF}-\frac{\pi }{2})}+\frac{\alpha e^{j\varphi_{\alpha }}}{2\sqrt{2}}\cdot V_{RF}\left[e^{j\left(\varphi_{RF}-\frac{\pi }{2}\right)}+e^{j\left(\varphi_{RF}-\pi \right)}\right]\cdot \Gamma_{BPF}$$
(3)$$V_{RFQ\_\alpha }=\frac{1}{\sqrt{2}}V_{RF}e^{j(\varphi_{RF}-\pi )}+\frac{\alpha e^{j\varphi_{\alpha }}}{2\sqrt{2}}\cdot V_{RF}\left[e^{j\left(\varphi_{RF}-\frac{\pi }{2}\right)}+e^{j\left(\varphi_{RF}-\pi \right)}\right]\cdot \Gamma_{BPF}$$


Assuming that both the Lange coupler and the power divider are ideal devices, it can be seen that the first terms in (2) and (3) are the RF signal input to the two mixing units, with equal amplitude and a 90° phase difference. The second item is the description of the process that the RF signal is coupled to the LO path and then reflected after reaching the BPF output port. Where α and $\varphi_{\alpha }$ are the amplitude and phase of the coupling, $\Gamma_{BPF}$ is the reflection coefficient of the BPF. Similarly, the process of LO signal coupling to the RF path and reflection at the output of LNA can also be described as follows:
(4)$$V_{LOI\_\alpha }=\frac{1}{\sqrt{2}}V_{LO}e^{j\varphi_{LO}}+\frac{\alpha e^{j\varphi_{\alpha }}}{2\sqrt{2}}\cdot V_{LO}e^{j\varphi_{LO}}\left(e^{j\left(-\frac{\pi }{2}\right)}+e^{j\left(-\pi \right)}\right)\cdot \Gamma_{LNA}\;e^{j\left(-\frac{\pi }{2}\right)}$$
(5)$$V_{LOQ\_\alpha }=\frac{1}{\sqrt{2}}V_{LO}e^{j\varphi_{LO}}+\frac{\alpha e^{j\varphi_{\alpha }}}{2\sqrt{2}}\cdot V_{LO}e^{j\varphi_{LO}}\left(e^{j\left(-\frac{\pi }{2}\right)}+e^{j\left(-\pi \right)}\right)\cdot \Gamma_{LNA}\;e^{j\left(-\pi \right)}$$


When Γ=0 or α=0, only the first term remains in (2)–(5); so, there are two straightforward solution methods. The first is to set the reflection coefficient to 0, which means that all LNA, mixer, and BPF need to be designed as broadband structures, and each port should match at both the RF frequency and the LO frequency. The second method is to reduce coupling by increasing the isolation of the mixing unit. These two methods both can reduce the influence on I/Q path imbalance. The second method was adopted in our design to fundamentally reduce the mutual coupling. Nevertheless, the degree of mutual coupling needs to be quantitatively weighed so that the amplitude and phase imbalances are acceptable.

In (2)–(5), the α is taken from −20 dB to −35 dB, and the phase is 0~360°. $V_{RFI\_\alpha }/V_{RFQ\_\alpha }$ is the imbalance of the RF signal caused by the coupling α. $V_{LOI\_\alpha }/V_{LOQ\_\alpha }$ is the imbalance of the LO signal. Assuming the extreme case, the ports are totally reflected (Γ=−1) and the RF and LO imbalances are superimposed. Figure 5 shows the amplitude and phase imbalance caused by the mutual coupling of the mixing unit. When the coupling α is less than −30 dB, the imbalance caused by it is approximately 0.35 dB and 2°.

The simplified diagram of the second coupling effect β is shown in Figure 6a. The 40 GHz LO coupling signal can enter into the mixing units through the Lange coupler and be superimposed with the LO signal from the power divider, which is expressed as follows:(6)$$V_{LOI\_\beta }=\frac{1}{\sqrt{2}}V_{LO}e^{j\varphi_{LO}}+\beta V_{LO}e^{j\left(\varphi_{LO}+\varphi_{\beta }\right)}\cdot \frac{1}{\sqrt{2}}e^{j\left(-\frac{\pi }{2}\right)}\;$$
(7)$$V_{LOQ\_\beta }=\frac{1}{\sqrt{2}}V_{LO}e^{j\varphi_{LO}}+\beta V_{LO}e^{j\left(\varphi_{LO}+\varphi_{\beta }\right)}\cdot \frac{1}{\sqrt{2}}e^{j\left(-\pi \right)}\;$$
where β represents the ratio of the coupling signal at the IRM input to the LO signal at the 40 GHz amplifier output, ranging from −20 dB to −35 dB. Calculate the imbalance of LO signal, and the results are shown in Figure 6b. When the coupling β is less than −28 dB, it will cause an imbalance of approximately 0.35 dB and 2°.

The third kind of coupling effect γ occurs in the crossover between the LO and the IF lines. As shown in Figure 7a, the 40 GHz LO signal will be coupled to the mixing unit on the Q path without affecting the I path:(8)$$\;\;\;V_{LOI\_\gamma }=\frac{1}{\sqrt{2}}V_{LO}e^{j\varphi_{LO}}$$
(9)$$V_{LOQ\_\gamma }=\frac{1}{\sqrt{2}}V_{LO}e^{j\varphi_{LO}}+\gamma V_{LO}e^{j\left(\varphi_{LO}+\varphi_{\gamma }\right)}$$

Similarly, the imbalance of the LO signal caused by coupling γ is calculated. The magnitude of γ ranges from −20 dB to −35 dB, and the phase ranges from 0 to 360°, as shown in Figure 7b. When the coupling is approximately −28 dB, the resulting imbalance is approximately 0.4 dB and 2°.

Although the influence of the above three kinds of coupling on the imbalance is of the same magnitude, in comparison, the coupling β and coupling γ are caused by the compact circuit layout in order to improve the high integration level, while the coupling α is due to the inherent characteristics of the mixer circuit topology, which is more critical and difficult to solve. Considering the simultaneous existence of the above three coupling effects, the amplitude and the phase imbalance caused by them may be superimposed or compensatory. Therefore, in order to satisfy the IRR > 20 dB of the integrated chip, the reasonable amplitude ranges of the three coupling effects are −25~−30 dB.

## 3. Coupling Effect Solution and Integrated Receiver MMIC Design

### 3.1. Solution to the Coupling in the Mixer RF-LO Path

Double balanced mixer (DBM) is a topology with high isolation characteristics, which is suitable for reducing mutual coupling in an RF-LO path. It provides a virtual ground by adding a balun to each port, so that all ports are mutually isolated. Furthermore, by using resistive FETs to realize DBM [13], the demand for LO power can be effectively reduced [15,16], which is also important for large-scale array applications to reduce power consumption. But relatively speaking, the design difficulty is increased, requiring additional bias on the gate of each FET.

In this design, the mixing unit consists of four 2 × 30 µm transistors in a ring configuration, where the LO signal is applied to the gate and the RF signal is applied to the source of the FET. The generated IF signal is extracted from the drain terminal. A three-conductor coupling line Marchand balun structure is adopted at the RF and the LO ports [17]. In the RF balun, capacitors are added to further reduce the circuit size [18]. Since the outer two conductors of the balun are connected together, the LO balun can also be employed as a circuit element to inject a bias voltage. For the IF balun with a relatively low frequency, in order to maintain a small circuit area, we adopt the spiral transformer structure [19]. On this basis, using the method of capacitive loading can also increase the coupling between the coils, and shorten the length of coupling lines. The schematic of a DBM unit is shown in Figure 8. For the IRM, besides the two mixing units described above, it also includes an RF quadrature-phase Lange coupler and an LO in-phase Wilkinson power divider. Considering that the Lange coupler has better broadband matching characteristics, it is more suitable for placement at the RF port, which requires a higher return loss than LO port.

When simulating the IRR, the IF output ports are connected to the ideal coupler. The in-band IRR is greater than 25 dB as shown in Figure 9a. The IRR simulation results reflect good amplitude and phase imbalances on the Lange coupler, power divider, and two mixing unit paths in the IRM design. Since multiple FETs and baluns are required to form a double-balanced mixing unit, it is more difficult to ensure the layout symmetry and matching of so many devices. Therefore, compared with single-ended mixers, more effort needs to be made in circuit design to obtain a satisfactory IRR for DBMs. In Figure 9b, the isolation of each port is greater than 34 dB, which shows the advantage of the DBM structure. It is also proved that the mutual coupling α existing in the mixer path is effectively reduced.

The results of the I/Q conversion gain varying with the 40 GHz LO power are shown in Figure 10a. If the gate bias is not added, the 40 GHz LO power needs to reach 16 dBm to make the IRM work normally. However, when the gate bias voltage VG = −0.7 V is injected, the LO power needs only to be greater than 5 dBm, which reflects the advantages of resistive FET. However, the mixing unit composed of resistive FETs is still a passive circuit, and the gate bias will not increase the conversion gain. Moreover, due to the existence of balun and Lange couplers in the double-balanced IRM, there will be an extra 6 dB loss compared to a single FET mixer. Figure 10b shows the result of an I/Q conversion gain in the normal working state of the IRM. In 32~36 GHz, the frequency conversion gain is −14~−15 dB, and the amplitude difference between the two I/Q paths is less than 0.5 dB.

According to the above analysis, the performance of IRM, especially IRR, is sensitive to the symmetry of circuit layout, so a Monte Carlo simulation is needed to analyze the sensitivity to process variations. The probability distribution functions of yield variables provided by the foundry are used for the simulation. Statistics are performed on each parameter as shown in Figure 11, where the ordinate is the proportion in 500-time simulations. It is shown that the variation range of the IRR is always between 24~29 dB, the RF-LO isolation is better than 34 dB. The range of conversion gain is −13~−15 dB, and the RF return loss is less than −18 dB. The Monte Carlo simulation results demonstrate that the performance varies within a predictable range.

### 3.2. Solution to the Coupling from Space and Dielectric

The LNA is composed of four 4 × 25 µm common-source transistors, with a gain of 22 dB in the range of 32~36 GHz, and 2.5 dB noise figure. The 40 GHz amplifier is a three-stage transistors structure with a gain of 16 dB. The layout of the two amplifiers and their positional relationship are shown in Figure 12. Since their transmission paths are parallel, the magnitude of the coupling β through space or dielectric is related to the distance d.

From Figure 13 a, it can be seen that S43 is the gain of the 40 GHz amplifier, and the coupling β=S23/S43. When d=0.8 mm, β is −20 dB. As the distance increases, the coupling decreases, but the area utilization of the chip needs to be considered. When d is finally determined to be 1.1mm, and isolated grounding vias are added, coupling β is −29 dB. To satisfy the drive power requirements of the mixer, the output power of the 40 GHz amplifier needs to be greater than 5 dBm. In this coupling state, in addition to the small signal received by the LNA, there will be a 40 GHz signal greater than −25 dBm in its transmission path, as shown in Figure 13b. When the coupling exists, the large signal in LNA is simulated by a harmonic balance simulation method. The results show that the gain of LNA remains unchanged, which is still 22 dB. The stability coefficient decreases slightly, but it is still far greater than 1. As in Figure 14, it verifies that the working state of the LNA is not affected by this coupling effect.

### 3.3. Solution to the Coupling Caused by the Line Intersection

It can be seen from the layout that in order to output the IF symmetrically from the right side of the chip, it is inevitable to cross over the LO transmission line. In MMIC design, air bridge structure is usually used to achieve crossover, but the microwave signals on the two lines are easy to be coupled, and when the transmission line is wider, the coupling is stronger. In Figure 15, two transmission lines with different widths are compared. When the LO line width at the crossover position is 70 um (characteristic impedance of 50 ohms), the coupling at 40 GHz is −15 dB. If the line width is narrowed to 12 um (characteristic impedance of 85 ohms), the coupling is reduced to −22 dB, and attention should be paid to the matching design of the circuit.

Considering the DBM circuit, Figure 16a shows the most intuitive way to use an air bridge on a single transmission line between the balun and the IF low-pass LC filter (L = 0.19 nH, C = 0.23 pF). Figure 16b shows the second method, where we modify the position of the air bridge to cross the LO transmission line on the differential line before the IF balun. In the two layouts, the coupling $\gamma_{1}$ of the LO to the DBM is not much different. However, the coupling $\gamma_{2}$ of the LO to the IF output port in the second structure can be further counteracted by the 180° reverse superposition of the IF balun. Suppressing the leakage of the LO power to off-chip is also meaningful for application in the system. The simulation results are shown in Figure 17. Coupling $\gamma_{1}$ is about −26 dB at 40 GHz, which meets the requirements for the influence of a coupling effect on IRR. The second structure significantly improves the suppression of coupling $\gamma_{2}$, so the differential line crossover is selected.

### 3.4. Implementation of the Integrated Receiver MMIC

Besides the LNA, IRM, and 40 GHz amplifier mentioned above, the quadrupler and phase shifter, 10 GHz amplifier, and quadrupler and BPFs in the integrated receiver MMIC are also designed separately.

The distributed analog phase shifter is a 4-stage cascade high-pass π-type network structure in which the variable capacitance is realized by varactors to change the insertion phase [20]. The 6 × 70 µm varactors are connected back-to-back, and control voltage (VC) fed to the cathode of the varactor through the parallel inductors. The schematic is shown in Figure 18a. When VC changes from 0 to 2 V, the phase shift range at 10 GHz is greater than 140°, and the insertion loss is 1.8 dB. The 10 GHz amplifier is a two-stage cascade structure composed of 2 × 45 um and 2 × 65 um FETs, and the gain reaches 23 dB. Its schematic is shown in Figure 18b. The single-FET quadrupler is the core of the LO circuit [21,22]. A 2 × 30 µm FET is selected with an operating bias of Vd = 2.2 V and Vg = −0.33 V to obtain maximum fourth-harmonic generation. At the gate of the transistor, open stubs TL1 and TL2 are used to short fourth and second harmonics to ground to avoid the interference of even harmonics. Meanwhile, two stubs TL3 and TL4 at drain are used to suppress the output of the second and third harmonics. Considering that the wavelength is too long for the fundamental frequency, lumped elements L3 and C7 are used instead of the stub. Figure 18c shows a schematic diagram of the quadrupler. Two BPFs are added at the output of the quadrupler and the 40 GHz amplifier to improve the suppression of harmonics. A three-stage structure with capacitive or inductive load coupling lines is used to realize narrow-band and small-sized filters [23], and the suppression of each harmonic is greater than 50 dB before moving into the IRM. The schematics of BPFs are shown in Figure 18d.

Figure 19 shows a microphotograph of the Ka-band receiver front-end MMIC based upon a standard 0.15 µm GaAs pseudomorphic high electron-mobility transistor process (pHEMT). As mentioned above, the chip includes an LNA, a double-balanced resistive FET image-reject mixer, a quadrupler, and a phase shifter. The chip size measures 5 × 4 mm^2^.

There are three kinds of DC ports: the +3 V drain bias of the LNA and the quadrupler, the −0.7 V gate bias of the mixer, and the 0 ~ +1.5 V control voltage of the phase shifter. The DC power consumption of the entire chip is only 330 mW.

The RF, LO, and IF pads are located at the three edges of the chip in the form of ground-signal-ground (GSG), which is easy to connect with external circuits through bonding wire.

## 4. Measurement Results and Comparison

The receiver MMIC was characterized using probed measurements. When measuring conversion gain vary with RF frequency, a 4-port Vector Network Analyzer (N5227B) scalar mixing mode is used. The instrument is not calibrated in this mode, and the measurement results are obtained by calculating the cable loss. The RF range is 30~38 GHz, and LO is fixed at 10 GHz, −13 dBm. The conversion gain measurement results of I/Q paths are shown in Figure 20a. The gain in the 32~36 GHz band is 6.1~8.7 dB, and the maximum amplitude difference between I/Q is approximately 0.7 dB. Although the measurement curves are not smooth, they are consistent with the trend of the simulation. When measuring the relationship between LO power and conversion gain, the RF is fixed at 34 GHz and the variation range of LO power is −25 to −5 dBm. As shown in Figure 20b, when the LO power is greater than −15 dBm, the chip is proper functioning. In addition, if the LO power continues to increase, the conversion gain remains constant. This result indicates that the output power of the LO chain is saturated in this range and drives the mixer to work properly. When applied to large-scale arrays, the receiver MMIC not only meets the requirements of low LO power but also maintains stability in a wide power range, which is conducive to the consistency of the system.

The return loss of each port is measured in the stable state and the results are shown in Figure 21. The RF return loss is worse than the simulation, but still less than −12 dB. The LO return loss is approximately −17 dB. The IF return loss is slightly higher at 8 GHz, but considering the application in the system, the next stage of the chip is connected with a well-matched IF hybrid. Therefore, the IF return loss of −8 dB is acceptable.

Noise figure is one of the most concerning parameters in a PMMI system, which is measured by a signal analyzer (FSVR40) and a noise source (NC346). From Figure 22, the measurement result of the average noise figure within the 32~36 GHz is less than 3.5 dB.

An Off-chip quadrature hybrid is used to combine IF signals, and the IRR is calculated by measuring the conversion gain in the RF and the image frequency. Figure 23 shows the IRR of the receiver MMIC, which is greater than 35 dB, reflecting the symmetric circuit design structure of the mixer and little impact of various coupling effects on the imbalance.

Since the receiver MMIC is a frequency conversion chip, the insertion phase cannot be measured directly, so another chip needs to be introduced as a reference. The change of phase with control voltage is shown in Figure 24. The measured phase shift range is smaller than that of the simulation, and it can be speculated that there is an error in the varactor model with the voltage variation. For this kind of cascade structure, the error is more obvious. However, when the control voltage changes from 0 to 1.5 V, the phase shift range is still >500°, meeting the system requirements of 360° continuous phase shift.

In general, the good measurement results of the receiver MMIC prove that the various coupling effects are effectively suppressed in the design of the integrated chip. The comparison with previously published receiver chips fabricated using a similar process is presented in Table 1. This shows that the reported receiver front-end MMIC not only has a higher integration but also certain advantages in noise figure and IRR, which is of more concern in PMMI applications.

## 5. Conclusions

This paper presents the design of a highly integrated Ka-band receiver front-end MMIC based on the GaAs process for a PMMI system. The coupling effects in integrated design are emphatically analyzed, and solutions are given for different coupling paths. The coupling has been effectively suppressed, all lower than −25 dB.

The noise figure of the fabricated MMIC is less than 3.5 dB, the conversion gain is better than 6 dB, and the IRR is greater than 35 dB. The DC power consumption is 330 mW and the LO power is only −15 dBm. Moreover, the phase can be continuously changed by more than 360°. These indicators show that the receiver MMIC can be used in the high-sensitivity large-scale array PMMI system.

## Figures and Tables

**Figure 1 sensors-22-05695-f001:**
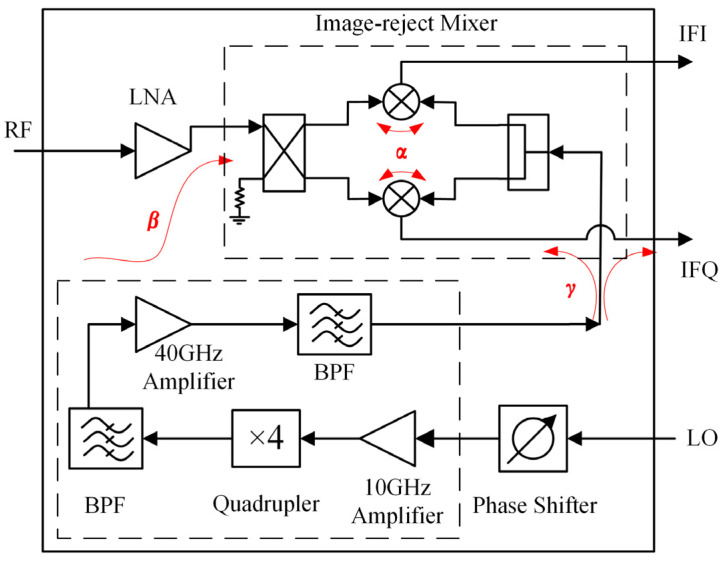
Block diagram of receiver MMIC and coupling paths.

**Figure 2 sensors-22-05695-f002:**
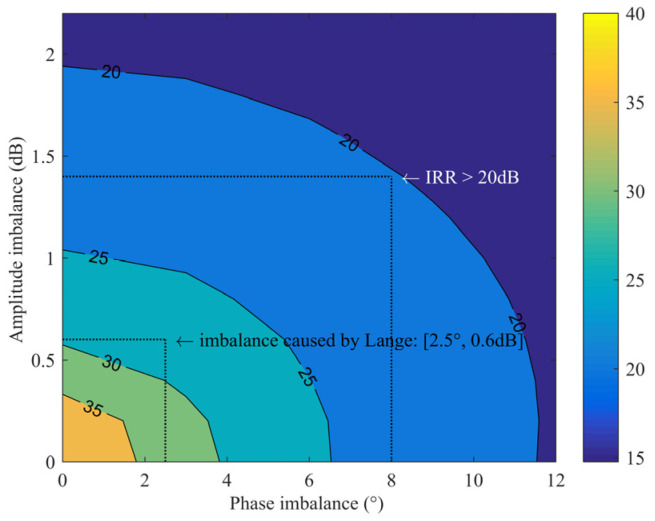
Relationship between the IRR and I/Q amplitude phase imbalance.

**Figure 3 sensors-22-05695-f003:**
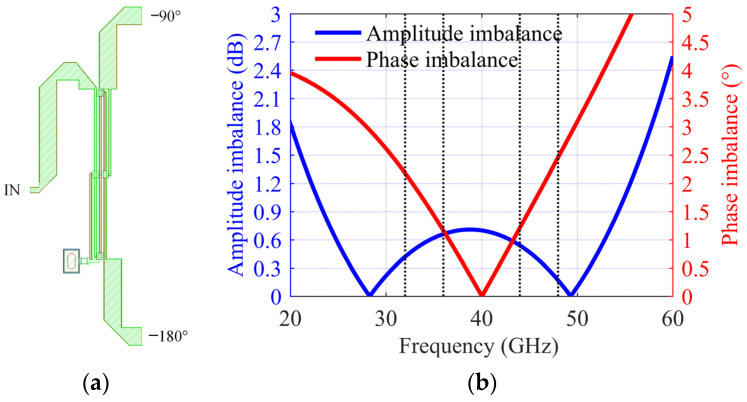
(**a**) Lange coupler simulation model; (**b**) Simulation results.

**Figure 4 sensors-22-05695-f004:**
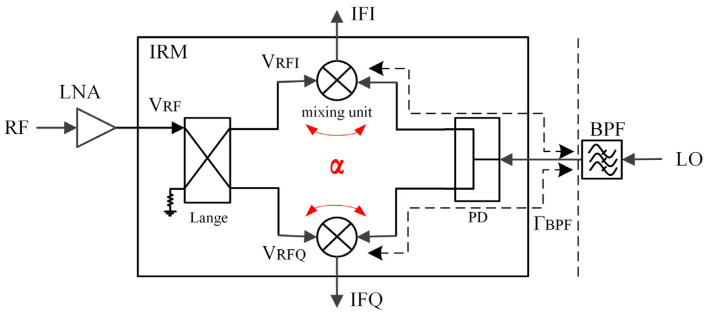
Analysis of the influence of RF-LO port mutual coupling α on the amplitude and phase imbalance.

**Figure 5 sensors-22-05695-f005:**
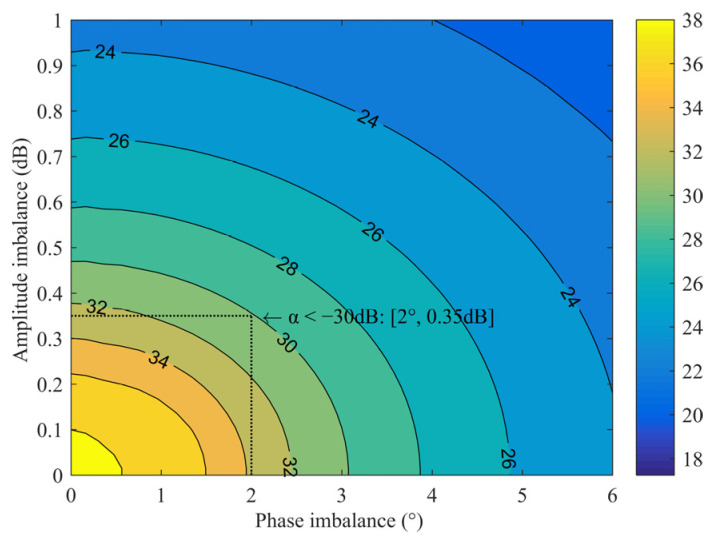
Amplitude and phase imbalance caused by the coupling α.

**Figure 6 sensors-22-05695-f006:**
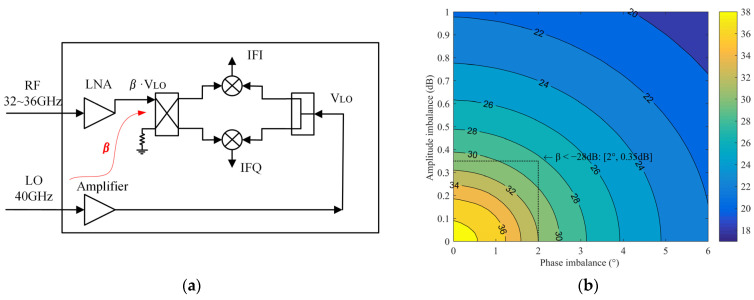
(**a**) Analysis of the influence of coupling β on the imbalance; (**b**) Amplitude and phase imbalance caused by the coupling β.

**Figure 7 sensors-22-05695-f007:**
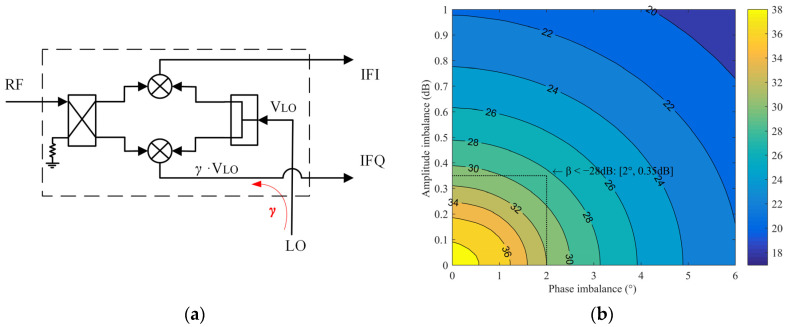
(**a**) Analysis of the influence of coupling γ on the imbalance; (**b**) Amplitude and phase imbalance caused by the coupling γ.

**Figure 8 sensors-22-05695-f008:**
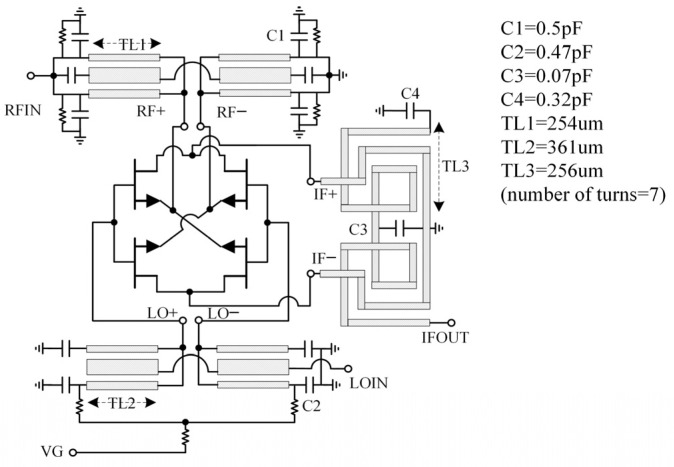
Circuit schematic of a double balanced mixing unit.

**Figure 9 sensors-22-05695-f009:**
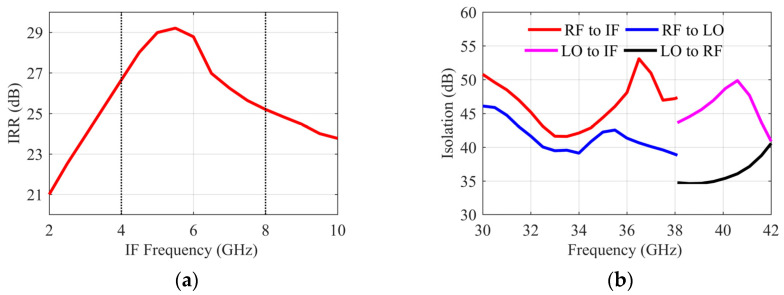
(**a**) Simulation result of IRR. (**b**) Simulation results of isolation between each port.

**Figure 10 sensors-22-05695-f010:**
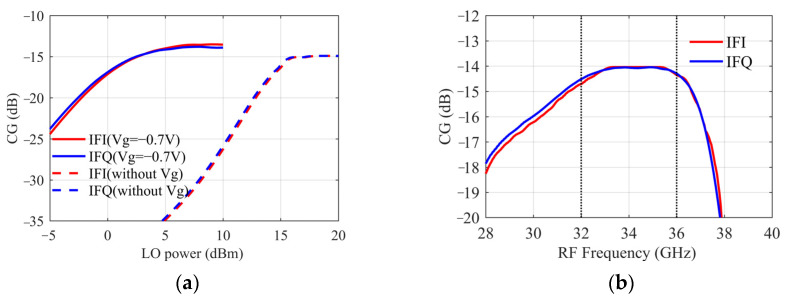
(**a**) Simulation result of conversion gain vs. LO power. (**b**) Simulation result of conversion gain vs. RF frequency.

**Figure 11 sensors-22-05695-f011:**
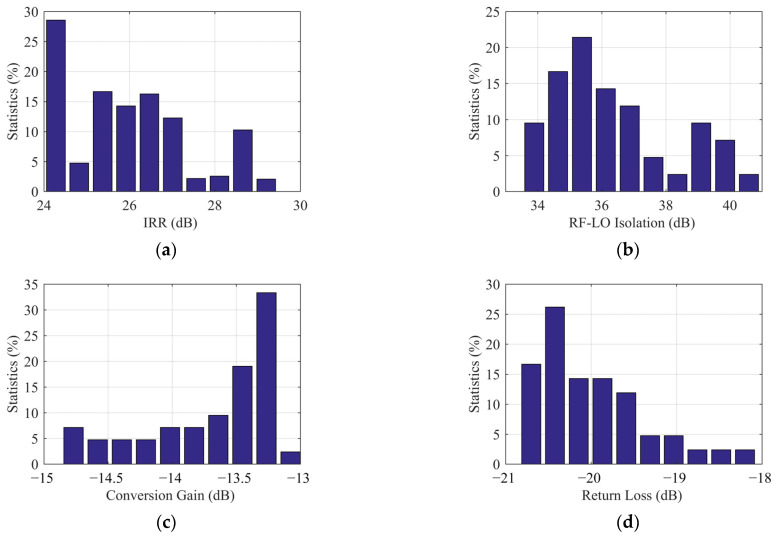
Statistical results of Monte Carlo simulation (**a**) IRR. (**b**) RF-LO isolation. (**c**) Conversion Gain. (**d**) RF Return Loss.

**Figure 12 sensors-22-05695-f012:**
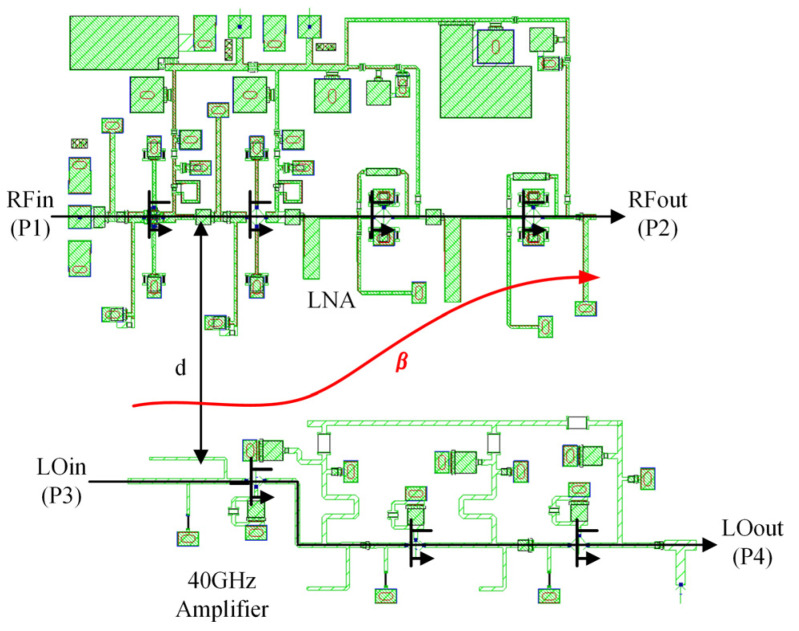
LNA and 40 GHz amplifier layout and their positional relationship.

**Figure 13 sensors-22-05695-f013:**
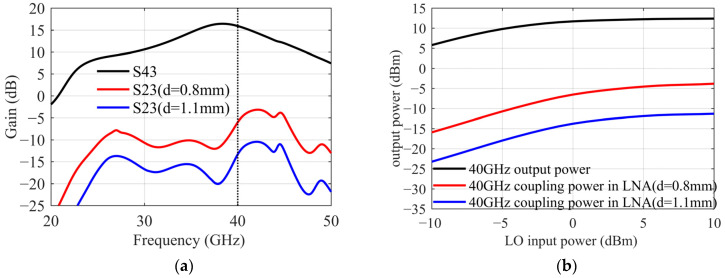
(**a**) 40 GHz amplifier gain and coupling. (**b**) 40 GHz amplifier output power and coupling power.

**Figure 14 sensors-22-05695-f014:**
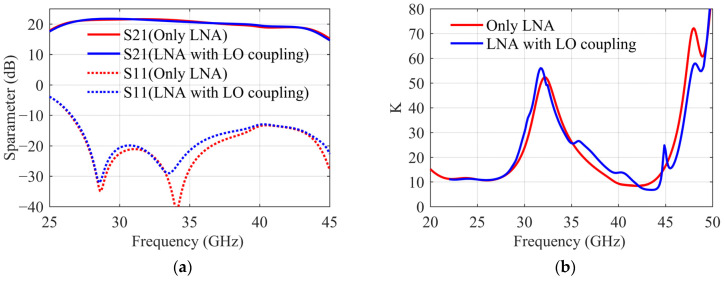
(**a**) LO coupling effect β on LNA S parameters. (**b**) LO coupling effect β on LNA stability factor.

**Figure 15 sensors-22-05695-f015:**
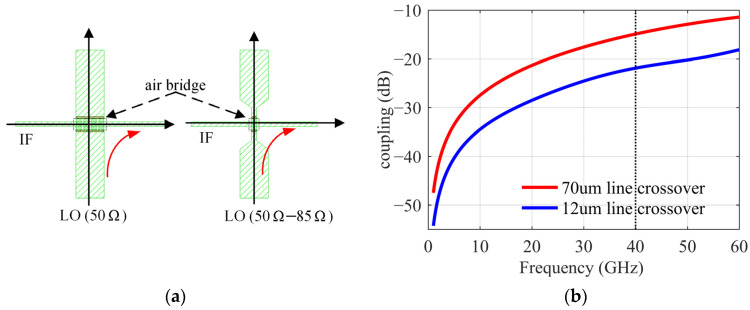
(**a**) Air bridge structure. (**b**) Comparison of two air bridge structures.

**Figure 16 sensors-22-05695-f016:**
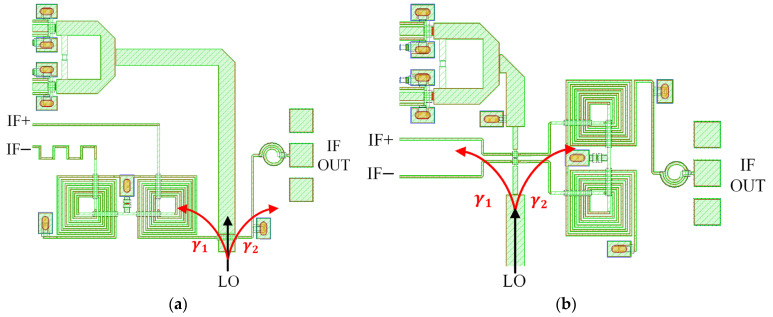
(**a**) The LO is directly crossed over the IF transmission line. (**b**) The LO is crossed over the IF differential lines.

**Figure 17 sensors-22-05695-f017:**
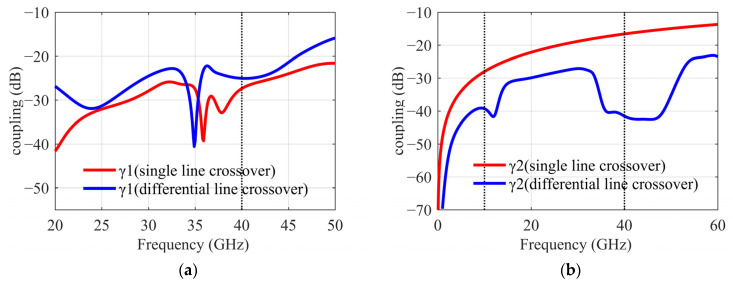
(**a**) Coupling $\gamma_{1}$ comparison of different crossover. (**b**) Coupling $\gamma_{2}$ comparison.

**Figure 18 sensors-22-05695-f018:**
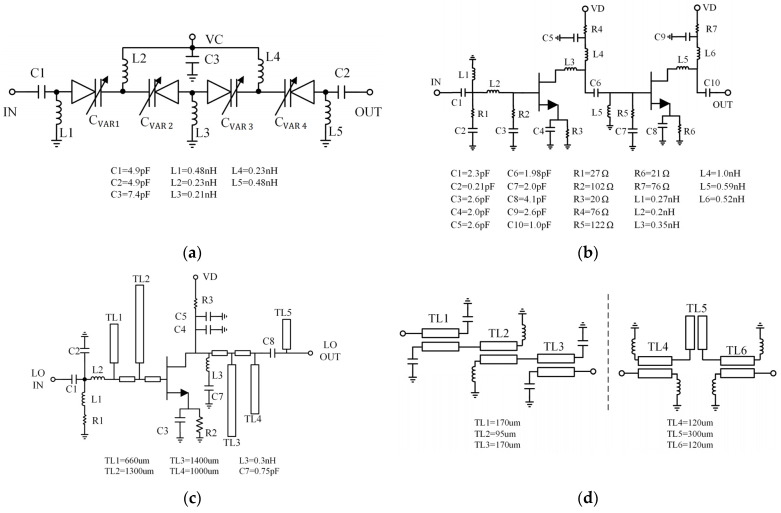
The schematic diagram. (**a**) Phase shifter. (**b**) 10 GHz amplifier. (**c**) Quadrupler. (**d**) BPFs.

**Figure 19 sensors-22-05695-f019:**
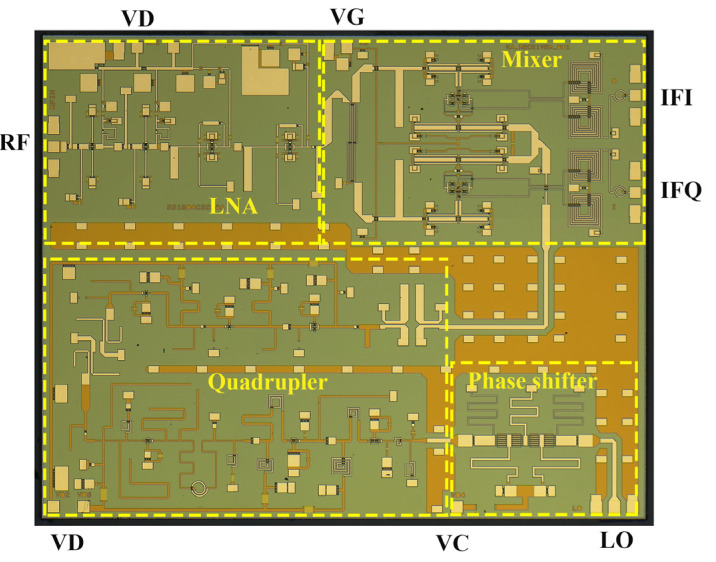
Microphotograph of the fabricated receiver MMIC. The chip size measures 5 × 4 mm^2^.

**Figure 20 sensors-22-05695-f020:**
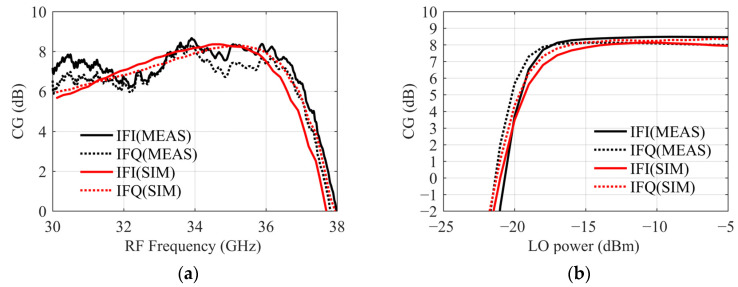
(**a**) I/Q conversion gain varies with RF frequency. (**b**) I/Q conversion gain varies with LO power.

**Figure 21 sensors-22-05695-f021:**
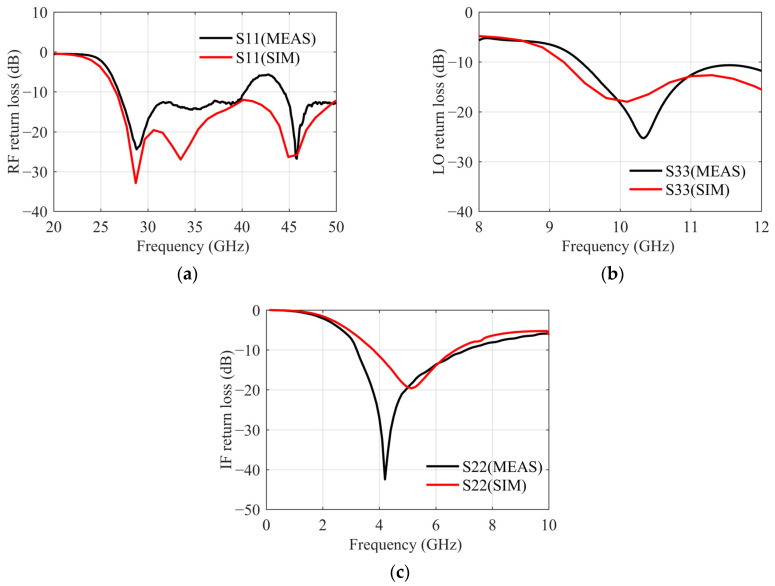
Return loss measurement results of each port. (**a**) RF return loss. (**b**) LO return loss. (**c**) IF return loss.

**Figure 22 sensors-22-05695-f022:**
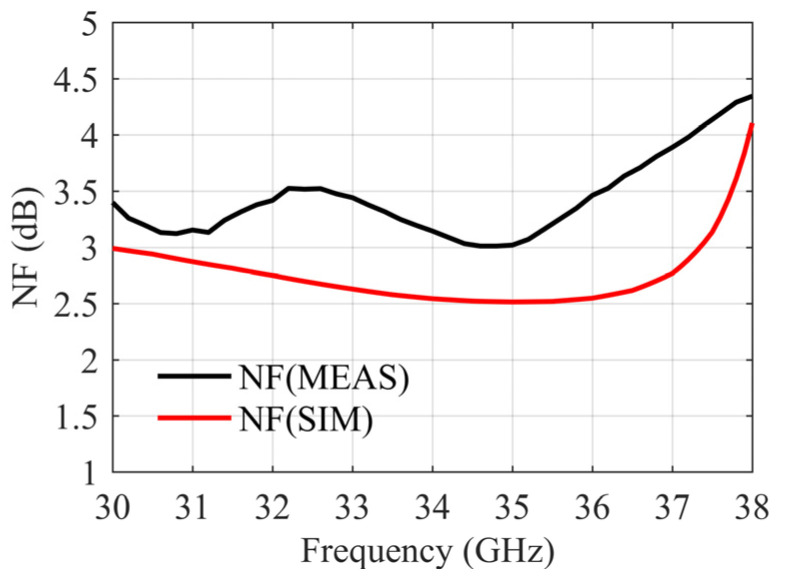
Noise figure results.

**Figure 23 sensors-22-05695-f023:**
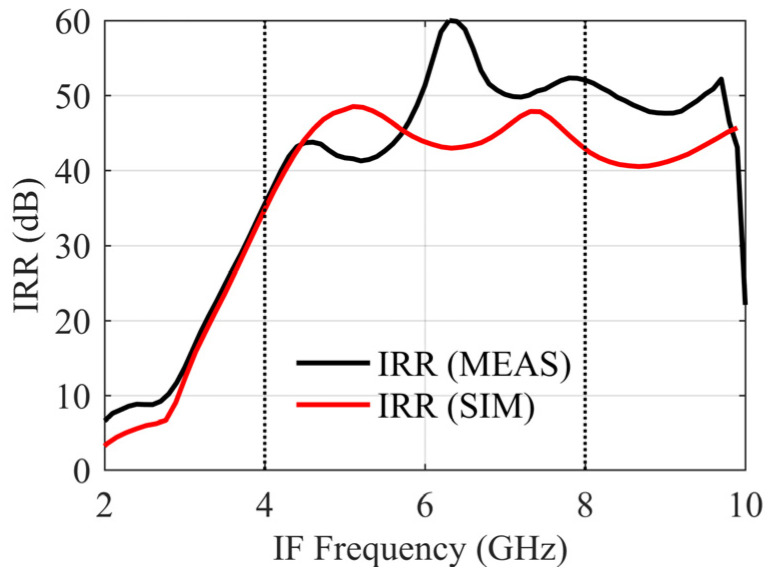
IRR result of receiver MMIC.

**Figure 24 sensors-22-05695-f024:**
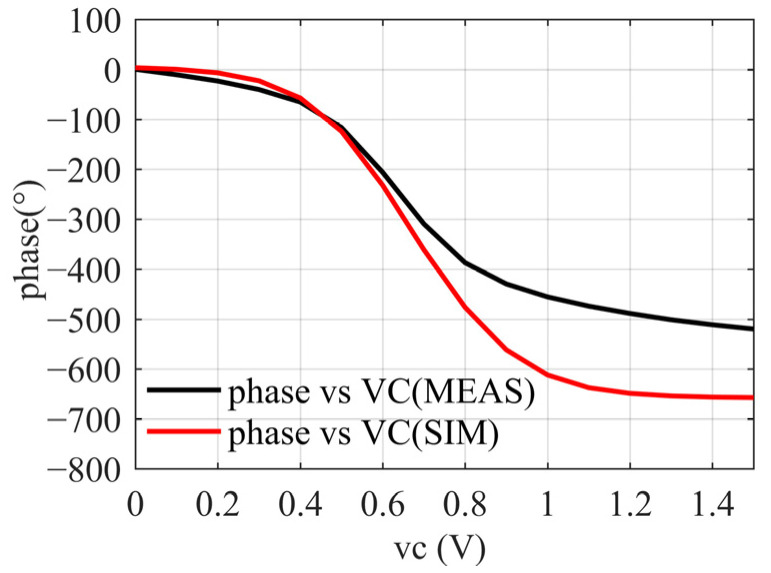
Results of phase shift vs. control voltage.

**Table 1 sensors-22-05695-t001:** Measured performance comparison of previously published receiver front-end MMICs.

Ref.	Technology	RF (GHz)	CG (dB)	NF (dB)	IRR (dB)	LO Power (dBm)	P_DC_ (mW)	Phase Shift Range (°)	Integration Level
[10]	0.15 µm GaAs pHEMT	29~36	10~14	3.5~4.5	20	9	800	-	LNA, sub-harmonic IRM
[11]	0.15 µm GaAs pHEMT	15~35	8	4.2	10	2	2000	-	LNA, nonbalanced IRM, LO buffer, LO doubler
[11]	0.15 µm GaAs pHEMT	35~45	10~12	3.5	20	2	800	-	LNA, IRM, LO buffer, LO doubler
[12]	0.15 µm GaAs mHEMT	56~64	10~15	7.2	13	−1	450	-	Amplifier, IRM, ×8 LO-chain
[24]	0.15 µm GaAs E-mode pHEMT	0.1~40	16	3.4~4.2	-	10	415	-	LNA, symmetric distributed drain mixer
[6]	45 nm CMOS	30~40	15	5	-	-	87.6	-	LNA, mixer, output buffer, LO buffer
[25]	28 nm CMOS	30–34.6	34.6	5.12	-	-	131	360	LNA, mixer, IF amplifier, LO buffer, phase shifter
This work	0.15 µm GaAs pHEMT	32~36	6.1~8.7	3~3.5	35	−15	330	500	LNA, double balanced resistive IRM, quadrupler, Phase shifter

## Data Availability

Not applicable.

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
