# Peer review of "Coupling Effects Analysis and Suppression in a Highly Integrated Ka-Band Receiver Front-End MMIC for a Passive Millimeter-Wave Imager System"

_sensors, 2022, doi:10.3390/s22155695_

Round 1

Reviewer 1 Report

This paper presents the coupling effects analysis and suppression of a highly integrated 11 receiver front-end MMIC for passive millimeter-wave imager system. The important coupling paths are modeled and simulated. The major problem is that the readers are confused about the basis of the model. And how to verify the accuracy of  these models. I think the authors should add some explanations about this. Another problem is that the references [10,11,12] in Table 1 are so old. The authors should compare with the latest references.

Reviewer 2 Report

Authors demonstrates highly integrated Ka-band receiver front-end with coupling effect analysis and supression, which can be applicable to passive millimeter-wave imager system. In this manuscript, authors handles various coupling scenarios with detailed simulations. My comments are listed in the following.

1. In this design, the receiver operates at 32-36 GHz band and IF ouput of 4~8 GHz is chosen. However, LO signal of 40 GHz is designed as high side injection. Is there specific reasons authors select 40 GHz LO signal instead of 28 GHz LO signal.  Normally, higher frequency LO signal requires more power consumption and lower output swing.

2. IRR of the mixer is also highly related to the device mismatch. I am curious the used process offers statistical data for Monte Carlo simulation. If included, the manuscrip will be more informative.

3. In Section 2.2, beta is defined as the coupling from the 40 GHz LO signal to the LNA from space or dielectric, affecting the performance of LNA. Howeverm, Fig. 6 and Eq. (6) and (7) depict LO coupling to input of mixer input unit (LNA output  From the perspective of coupling, the coupling from LO to LNA input is more critical because leaked LO signal is also amplified by LNA. Please add the description and clarify the definition of couplig factor beta.

4. In section 3.3, please include the detailed dimensions of the implemented Marchand balun. Please also include design parameters of utilized output inductors.

5. In the mixer design, the DBM is a common approach to effectively enhance RF-LO port isolation. However, I doubt that DBM has own advantage IRR. Please incldue more clear description regarding this point.

6. mixer conversion gain is relatively low ( -14~-15 dB). please add related description

7. In the manuscript, the schematic of each block is not included. Please add block schematics of LO amplifer, phase shifter and filters.

Reviewer 3 Report

This paper studied the coupling effects of receiver front-end MMIC system. It is interesting for MMIC. Three different paths is studied on IRR. I have some comments.

1. Figure 9,10,12,13,16,18,19,20,21 : freq should be frequency.

2. section 3.4: Where is the varactor? Please provide the circuit schematic of phase shifter.

3. The authors provide three paths affect coupling. What is the most important for these three paths?

4. How to choose between area and path?

5. Fig. 1 show block diagram for Fig. 17. Only three coupling paths. How about phase shifter to Quadrupler ?
